# Inequalities in care for the people with diabetes in Brazil: A nationwide study, 2019

**Rosália Garcia Neves** [1]*, **Mirelle de Oliveira Saes**[2], **Suele Manjourany Silva Duro**[3], **Thaynã Ramos Flores**[4], **Elaine Tomasi**[4]

**1** State Department of Health, Rio Grande do Sul, Porto Alegre, Brazil, **2** Postgraduate Programme in Health Sciences, Federal University of Rio Grande, Rio Grande, Brazil, **3** Postgraduate Programme in Nursing, Federal University of Pelotas, Pelotas, Brazil, **4** Postgraduate Programme in Epidemiology, Federal University of Pelotas, Pelotas, Brazil

* rosaliagarcianeves@gmail.com

**Data Availability Statement:** All files are available from database https://www.ibge.gov.br/estatisticas/downloads-estatisticas.html?caminho=PNS/2019/Microdados/Dados.

## Abstract

The purpose of this paper is to evaluate inequalities in care for people with diabetes in Brazil. This cross-sectional population-based study was carried out in 2019 and evaluated care provided by receiving advice, requesting laboratory tests, and performing examinations. We used the slope index of inequality and concentration index to assess inequalities according to educational level and Poisson regression to estimate prevalence ratios for each outcome in the education category. We assessed a total of 6317 people with diabetes, 41.8% had their eyes checked, and 36.1% had their feet examined in the previous year. Prevalence for both examinations was 2.45 times higher in those from the highest level of education compared to those from the lowest level. The largest absolute differences (in percentage points) between the lowest and highest education levels in care indicators were the following: request for glycated hemoglobin test (39.0), glucose curve test (31.4), and eyes checked in the previous year (29.7). There were notable inequalities in the prevalence ratios of care provided to people with diabetes in Brazil. Requests for glycated hemoglobin tests, glucose curve tests, eye and feet examinations should be emphasized, especially for people from lower educational levels.

## Introduction

Diabetes mellitus (DM) represents a major cause of morbidity and mortality in the population, ranking 5th in assessing disability-adjusted life years (DALYs) worldwide in 2019 [1]. The global prevalence of diabetes was approximately 9% [2]. It rose from 6% in 2013 to 8% in Brazil in 2019, reaching 13% among people from low socioeconomic levels [3]. Various complications are linked to diabetes, such as kidney disease, amputations, blindness, increased risk of other cardiovascular diseases and stroke, leading to adverse health outcomes, especially hospitalizations, disability, and deaths [4, 5].

Controlling DM, as a Chronic Non-Communicable Disease (NCD), depends both on self-care and the health system, with emphasis on primary care for case management and monitoring to prevent complications [6]. Such control should be reinforced among the most vulnerable population, which is predominant in Brazil, where poverty levels have been increasing in recent years [7].

**Funding:** The author(s) received no specific funding for this work.

**Competing interests:** The authors have declared that no competing interests exist.

DM disproportionately affects the lowest socioeconomic groups, reinforcing the direct relationship between health and economic conditions [8, 9]. Although Brazil has a universal, comprehensive and equitable Unified Health System with Primary Health Care responsible for the management of DM, the quality of care received, especially by people with lower socioeconomic status, is affected by infrastructure, lack of access to services, health professional work process, and the population's socio-demographic characteristics, contributing to poorer health outcomes [6, 10–13]. On the other hand, people with higher socioeconomic status, in the country, tend to use the private health system, being able to obtain greater access and quality in every line of care for the people with DM [4, 8, 9, 12].

Regarding the quality of care offered to people with diabetes, most studies in Brazil have evaluated access and structure, while a gap remains considering the work process quality evaluation [10, 11]. Similarly to this study, Neves et al. [13] assessed social inequities in the care of older people with diabetes based on data from the 2013 National Health Survey (PNS), and identified worse care for the poorest. This result was found mainly for advice on measuring blood glucose, requests for glycated hemoglobin tests, requests for glucose curve tests and examination of the eyes and feet, resulting in low performance of all evaluated care services (one in ten older people).

The objective of this article is to evaluate inequalities in care for people with diabetes in Brazil. The absence of studies between 2013 and 2019 on this topic, and the opportunity to monitor the situation of the care received in the health system in the period of six years may bridge an existing gap in the literature, and increase policy efforts on the health of people with DM.

## Methods

This cross-sectional population-based study uses data from the National Health Survey (PNS), open access, carried out in Brazil in 2019 by the Brazilian Institute of Geography and Statistics (IBGE) in partnership with the Ministry of Health. The sample was representative of permanent residents living in urban or rural areas of municipalities in Brazil's five geographic regions, distributed over 26 Federative Units and Federal District.

The sampling process was done in three stages. First, census tracts were selected, then households, and finally, individuals aged 18 or older. The sample was made up of 108,457 households, where 90,846 individuals answered the questionnaire on chronic diseases.

Trained interviewers collected data using handheld computers (personal digital assistant [PDA]) for data storage. The questionnaire consisted of three parts: a) household variables, b) general characteristics of all residents in the household, and c) work and health related questions asked to one randomly selected resident. The present study sample consisted of adults aged 18 years or older who reported a medical diagnosis of DM and who had seen a doctor due to diabetes in the previous three years. More details about the sampling process and instruments are available in the PNS document [3].

In order to evaluate the care offered to people with DM, three synthetic outcomes were created using the collected information: 1) Receiving all types of advice, based on questions about having a healthy diet, maintaining adequate weight, practicing regular physical activity, not smoking, not drinking in excess, reducing consumption of pasta and bread, avoiding consumption of sugar, sugary and sweet drinks, measuring blood glucose at home, examining feet regularly and having regular monitoring with a health professional, based on the following question: "In any of your diabetes consultations, did any doctor or other health professional give you any of these recommendations?"; 2) Requesting all tests, including blood glucose, glycated hemoglobin, glucose curve, urine analysis, and cholesterol or triglycerides, using the following question: " Were any of these tests requested in your diabetes consultations?" and 3)

Eye and feet examinations in the previous year, by a health professional, based on the following questions: "When was the last time you had your eyes checked in which your pupil was dilated?" and "When was the last time a doctor or health care professional examined your feet for sensitivity or the presence of sores or irritation?"

The independent variable was education divided into five categories (no education; incomplete elementary school; complete elementary school / incomplete high school; complete high school / incomplete higher education and complete higher education) and the variables used for adjustment were region (North; Northeast; Midwest; Southeast; South), sex (male; female), age in complete years (18 to 49; 50 to 64; 65 and over) and self-reported skin color (white; black; brown/yellow/indigenous).

We calculated prevalence and 95% confidence intervals (CIs) for each of the care indicators and performed an adjusted analysis using Poisson regression to estimate the prevalence ratios and the respective confidence intervals according to education categories.

We estimated the magnitude of the inequalities for each indicator regarding the education variable using two indices: the slope index of inequality (SII) and the concentration index (CIX). The SII expresses absolute difference, in percentage points, between the prevalence of extreme education categories, using a logistic regression model. The CIX is based on a scale ranging from -100 to +100, with zero representing equal distribution across schooling categories, while positive values indicate that the distribution favors the most educated. The SII presents absolute inequality, while the CIX indicates relative inequality. We calculated 95% CIs for the SII and CIX. Several authors currently use these indices to measure health inequalities [14–16]. More details of the analyses can be found in Silva et al. [17]. We performed all analyses using STATA® 15.0 statistical package, using the "svy" command. That command takes into account the survey design, including sampling weights of the individual and clustering.

The National Research Ethics Committee of the National Health Council approved the National Health Survey project in August 2019 under protocol number 3.529.376. All participants signed a free and informed consent form, and ethical principles were safeguarded.

## Results

Of the 90,846 respondents, 7,358 individuals reported a previous medical diagnosis of DM (8.1%). From these, 6,317 (85.9%) had received medical care in the previous three years, making up the sample of the present study. Approximately half of the sample was located in the Southeast (49.4%), 57.0% were female, 41.7% were 65 or over, 45.2% reported white skin color, and most had incomplete primary education (46.2%) (Table 1).

Fig 1 shows the prevalence for each of the studied indicators. The most prevalent types of advice were to keep a healthy diet (95.0%), avoid sugar consumption (92.9%), and maintain adequate weight (92.1%). On the other hand, advice on measuring blood glucose (65.9%) and examining the feet (53.3%) was the least prevalent. Receiving all types of advice was reported by 32.9% of respondents. Blood glucose and triglycerides were the most frequently required laboratory test, 93.3% and 86.8% respectively, and the glucose curve was the least requested (54.1%). All tests were found to be requested for 45.0% of the sample. Less than half had their eyes (41.8%) and feet examined (36.1%) in the previous year, and 21.0% underwent both examinations.

Concerning the analysis of inequalities for most indicators, the highest proportions were found in the highest schooling categories (complete high school/ incomplete higher education and complete higher education). Advice on alcohol consumption, glycated hemoglobin, glucose curve, triglycerides test requests, and eye examination in the previous year and performance of all tests showed higher prevalence as the level of education increased (Fig 2).

**Table 1. Description of the sample according to regional and sociodemographic characteristics in the people with diabetes mellitus, Brazil, 2019 (N = 6317).**

| Variable | N (%)* |
|---|---|
| Region | |
| North | 929 (5.3) |
| Northeast | 2.139 (23.9) |
| Midwest | 738 (6.6) |
| Southeast | 1.681 (49.4) |
| South | 830 (14.8) |
| Sex | |
| Male | 2.512 (43.0) |
| Female | 3.805 (57.0) |
| Age (years) | |
| 18–49 | 978 (17.3) |
| 50–64 | 2.527 (41.0) |
| 65 and over | 2.812 (41.7) |
| Skin color | |
| White | 2.415 (45.2) |
| Black | 787 (11.6) |
| Brown/yellow/indigenous | 3.115 (43.2) |
| Educational level | |
| No education | 1.000 (12.5) |
| Elementary education incomplete | 2.803 (46.2) |
| Elementary education completed/high education incomplete | 701 (11.3) |
| High education completed/higher education incomplete | 1.205 (20.6) |
| Higher education completed | 608 (9.4) |

*These are the absolute number and the weighted sample proportion

Table 2 describes the reasons for outcome prevalence according to the exposure variable. Advice on practicing physical activity and advice not drinking too much was about 20% more prevalent in individuals with higher education level than those who had no education. Receiving all types of advice was approximately 35% more prevalent among those who had complete high school education or above. Requesting glycated hemoglobin tests, glucose curve tests, and all tests showed a positive association since there was an increase in this prevalence as education level increased. Having had an eye examination in the previous year was 1.74 times higher among those with complete higher education compared to those with no education, and feet examination was 1.45 times higher. Prevalence for both examinations was 2.45 times higher in those with a higher level of education compared to those from a lower level.

Among the evaluated indicators, four showed the greatest absolute differences represented by the SII: request for glycated hemoglobin tests (39.0p.p.), glucose curve tests (31.4p.p.), eyes examined in the previous year (29.7p.p.) and all requested tests (29.0p.p.). The relative inequalities (CIX) were greater for the indicators requesting all laboratory tests, eyes examined in the last year and performance of all examinations (Table 3).

## Discussion

We identified inequalities in care for individuals with DM. Those from higher schooling levels were more likely to receive complete advice on the management of DM, have all tests requested by health professionals, and perform all exams. We found that the probability of

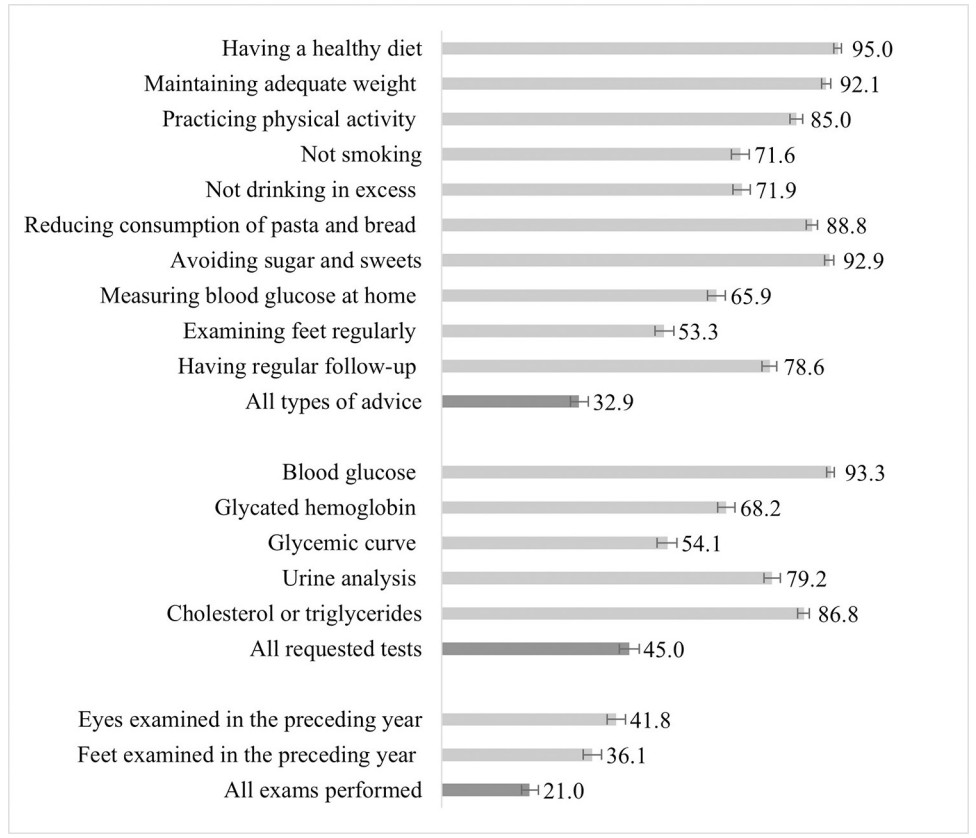

**Fig 1. Prevalence (%) of the care services offered to the people with diabetes mellitus, Brazil, 2019, (N = 6317).**

having all exams was two times greater among people with higher education than those with no education. The greater the quality of care, the greater the difference between education categories, with emphasis on the most educated. This finding corroborates with Neves et al. [1] indicating the persistence of inequities in the quality of care for patients with DM after six years, and reinforcing the existence of gaps in the qualification of clinical care and access to tests and exams, especially in Primary Care among individuals aged 60 years or older, when evaluating similar indicators from the 2013 PNS. In addition, our findings corroborate with results found in the literature for Latin American countries where inequalities in health interventions are evident [18].

The quality of care provided to patients with DM can influence the evolution of other diseases related to it [5, 6]. About a third of the sample reported having received all types of advice evaluated, consisting of cost-free actions to be carried out in all contacts between health professionals and their patients, providing information and education [19, 20]. The importance of receiving advice from health professionals should be emphasized so these habits are effectively put into practice [21].

The use of educational level as an exposure variable in the present study can strengthen our findings, considering that some types of advice are possibly widely known and not necessarily provided by health professionals who monitor patients [22]. However, it should be emphasized that low education levels can impair patients' understanding of the disease and its treatment, and undermine the importance of self-care [23].

Advice on measuring blood glucose and advice on examining feet were the least prevalent. Several studies have shown the same problem regarding the quality of care received in Primary

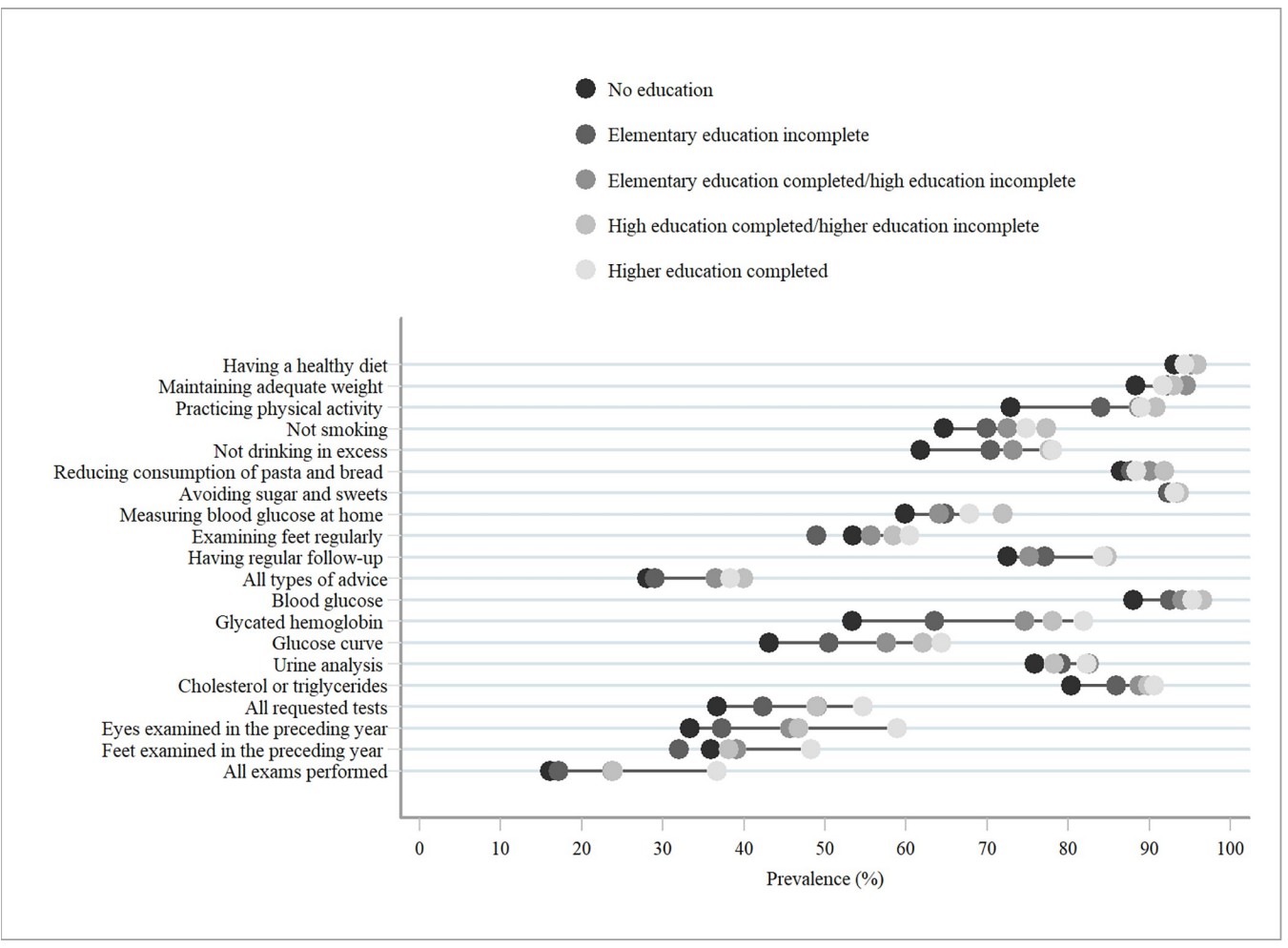

**Fig 2. Prevalence (%) of the care services offered to the people with diabetes mellitus, according to educational level, Brazil, 2019, (N = 6317).**

Health Care (PHC). A study with 8,118 PHC users linked to family health teams and a medical diagnosis of DM found that only 49% received guidance on foot care [10] and Santos et al. [24] found a 35% prevalence of receiving this type of guidance. Gonçalves et al. [25], in Porto Alegre/RS, evaluated the prevalence of different types of advice among users of services with high and low general PHC scores of quality of care, according to the PCATool, and found a difference around two times higher for advice on feet examination and 1.3 times higher for advice on healthy eating in high-score services.

The report on a foot care task force conducted by the *American Diabetes Association* stressed the importance of health service users with DM having their feet assessed at least once a year and recognizing signs of possible complications to reduce lower limb amputations [26]. Batista et al. [27] showed that low education levels hinder appropriate feet care, mainly due to reduced understanding of the disease and guidance given. For this reason, this type of advice should be clearly and objectively given.

Less than a half of studied population had performed all recommended tests during the 12 month period. The glucose curve was the least requested test, and the glycated hemoglobin test was requested for less than 70% of the sample. Laboratory tests are essential for monitoring these measurements to achieve qualified clinical management and control of the disease.

**Table 2. Adjusted analysis of the care services offered to the people with diabetes mellitus, according to educational level, Brazil, 2019 (N = 6317).**

| Variable | Educational level [Prevalence Ratio (95% CI)] | | | | |
|---|---|---|---|---|---|
| | No education | Elementary education incomplete | Elementary education completed/ high education incomplete | High education completed/higher education incomplete | Higher education completed |
| Having a healthy diet | 1.00 | 1.01 (0.98.1.04) | 1.01 (0.98.1.05) | 1.02 (0.98.1.05) | 1.00 (0.96.1.04) |
| Maintaining adequate weight | 1.00 | 1.04 (1.00.1.08) | **1.06** (1.02.1.11) | 1.04 (1.00.1.09) | 1.03 (0.98.1.08) |
| Practicing physical activity | 1.00 | **1.14** (1.06.1.22) | **1.19** (1.10.1.29) | **1.21** (1.12.1.31) | **1.20** (1.11.1.30) |
| Not smoking | 1.00 | 1.06 (0.96.1.17) | 1.08 (0.97.1.21) | **1.15** (1.03.1.27) | 1.12 (0.99.1.26) |
| Not drinking in excess | 1.00 | 1.10 (1.00.1.22) | **1.13** (1.01.1.27) | **1.18** (1.06.1.31) | **1.19** (1.06.1.34) |
| Reducing consumption of pasta and bread | 1.00 | 0.99 (0.95.1.04) | 1.01 (0.96.1.07) | 1.02 (0.97.1.08) | 1.00 (0.94.1.06) |
| Avoiding sugar and sweets | 1.00 | 0.99 (0.96.1.02) | 1.00 (0.96.1.04) | 1.00 (0.96.1.04) | 1.00 (0.96.1.04) |
| Measuring blood glucose at home | 1.00 | 1.07 (0.96.1.19) | 1.05 (0.93.1.19) | **1.18** (1.05.1.32) | 1.11 (0.98.1.27) |
| Examining feet regularly | 1.00 | 0.93 (0.82.1.05) | 1.06 (0.90.1.25) | 1.11 (0.97.1.28) | 1.14 (0.98.1.32) |
| Having regular follow-up | 1.00 | 1.04 (0.97.1.12) | 1.01 (0.91.1.12) | **1.13** (1.05.1.22) | **1.14** (1.04.1.24) |
| **All types of advice** | 1.00 | 1.02 (0.83.1.24) | 1.27 (1.00.1.61) | **1.37** (1.10.1.71) | **1.34** (1.05.1.71) |
| Blood glucose | 1.00 | 1.04 (1.00.1.08) | **1.05** (1.01.1.10) | **1.08** (1.04.1.12) | **1.07** (1.01.1.12) |
| Glycated hemoglobin | 1.00 | **1.15** (1.03.1.29) | **1.36** (1.20.1.53) | **1.44** (1.28.1.62) | **1.49** (1.32.1.68) |
| Glucose curve | 1.00 | 1.12 (0.97.1.30) | **1.29** (1.09.1.53) | **1.42** (1.21.1.66) | **1.44** (1.20.1.71) |
| Urine analysis | 1.00 | 1.05 (0.98.1.12) | **1.10** (1.02.1.19) | 1.05 (0.97.1.14) | **1.10** (1.01.1.20) |
| Cholesterol or triglycerides | 1.00 | 1.06 (1.00–1.12) | **1.10** (1.03.1.17) | **1.11** (1.05.1.18) | **1.12** (1.04.1.20) |
| **All requested tests** | 1.00 | 1.14 (0.96–1.37) | **1.34** (1.10–1.64) | **1.37** (1.13–1.67) | **1.48** (1.20–1.83) |
| Eyes examined in the preceding year | 1.00 | 1.12 (0.94–1.34) | **1.39** (1.11–1.75) | **1.44** (1.19–1.75) | **1.74** (1.43–2.13) |
| Feet examined in the preceding year | 1.00 | 0.94 (0.78–1.14) | 1.20 (0.95–1.52) | 1.23 (0.99–1.52) | **1.45** (1.17–1.80) |
| **All exams performed** | 1.00 | 1.14 (0.84–1.56) | **1.66** (1.15–2.40) | **1.75** (1.26–2.43) | **2.45** (1.74–3.44) |

CI: Confidence Interval

Studies [28, 29] have shown that maintaining glycated hemoglobin (HbA1c) levels below 7% can decrease vascular complications of diabetes and that the higher the levels of glycated hemoglobin, the greater the severity of diabetic neuropathy.

The prevalence of examining the feet and eyes in the previous year can be considered low. Corroborating with our findings, Tomasi et al. [10] found 46% occurrence of the fundus examination performed periodically, and only 33% of feet checked among primary care users. Similarly, a study carried out with patients hospitalized with diabetic foot found a 44% prevalence of having their feet examined during routine consultations in the previous year [24]. When comparing other Latin American countries, Gagliardino et al. observed that a little more than one-third and around eight in ten participants had their eyes and feet checked, respectively [30].

In addition, we found that only 21% of the respondents had had both of these exams, with the aggravating factor that the highest occurrence was among the most educated. These tests can and should be performed during routine appointments, as recommended by national and international guidelines for disease control [2, 6, 31], and are important indicators of quality of care for individuals with DM and preventing the onset of disabilities and irreversible blindness [32].

**Table 3. Slope index of inequality and concentration index, with 95% confidence intervals, of the care services offered to the people with diabetes mellitus, according to educational level, Brazil, 2019 (N = 6317).**

| Variable | Slope Index of Inequality | 95% CI* | Concentration Index | 95% CI* |
|---|---|---|---|---|
| Having a healthy diet | 3.9 | 1.8–6.0 | 0.5 | 0.2–0.8 |
| Maintaining adequate weight | 7.7 | 5.1–10.4 | 1.3 | 0.9–1.7 |
| Practicing physical activity | 19.8 | 16.5–23.1 | 3.4 | 2.8–4 |
| Not smoking | 10.5 | 6.5–14.5 | 2 | 1.1–2.9 |
| Not drinking in excess | 15.4 | 11.4–19.4 | 3.4 | 2.5–4.3 |
| Reducing consumption of pasta and bread | 5.1 | 2.2–7.9 | 0.8 | 0.3–1.3 |
| Avoiding sugar and sweets | 3.2 | 0.9–5.6 | 0.5 | 1.1–3.2 |
| Measuring blood glucose at home | 10.0 | 5.8–14.2 | 2.1 | 1.1–3.2 |
| Examining feet regularly | 14.4 | 10.0–18.8 | 4.3 | 3.0–5.7 |
| Having regular follow-up | 14.0 | 10.2–17.8 | 2.8 | 2.0–3.6 |
| **All types of advice** | **14.0** | **10.0–18.4** | **6.1** | **4.1–8.1** |
| Blood glucose | 10.5 | 7.9–13.2 | 1.6 | 1.2–2.0 |
| Glycated hemoglobin | 39.0 | 35.2–42.9 | 9.0 | 8.0–10 |
| Glucose curve | 31.4 | 27.2–35.5 | 9.0 | 7.7–10.4 |
| Urine analysis | 12.1 | 8.5–15.8 | 2.1 | 1.4–2.9 |
| Cholesterol or triglycerides | 16.7 | 13.3–20.0 | 3.0 | 2.4–3.6 |
| **All requested tests** | **29.0** | **24.8–33.2** | **9.6** | **8.0–11.2** |
| Eyes examined in the preceding year | 29.7 | 25.6–33.9 | 11.7 | 9.9–13.4 |
| Feet examined in the preceding year | 15.3 | 11.0–19.5 | 6.4 | 4.5–8.4 |
| **All exams performed** | **19.5** | **15.8–23.1** | **14.6** | **11.8–17.4** |

*CI: Confidence Interval

Another point to be considered is that according to the literature, the less educated tend to consult more in primary health care services. These services have historically had worse infrastructure [11, 12]. Neves et al. [11] found that, from the primary care teams in Brazil, only 31% had a monofilament kit. Only 23% had an ophthalmoscope available at their primary care centers, and less than 8% of the teams had an adequate minimum structure of materials to care for people with DM.

Recall bias stands out as a limitation of this study. This bias was identified by the absence of specific temporality for questions related to guidance received. In this period, individuals may have had more opportunities to receive any type of evaluated advice or even be confused with some other moment in life when they received such recommendations, so that the estimates found may have been overestimated. We believe that temporality, such as 12 months prior to the interview for eye and foot examinations, could have minimized this limitation.

It is noteworthy that, in view of the possibility of using schooling as a proxy for socioeconomic level, a correlation test was carried out between schooling and wealth index. This test revealed a high relationship (0.8) between the variables, which contributed to the use of schooling in the analysis of inequalities in care for individuals with DM.

Absolute inequality and relative inequality (expressed by SII and CIX, respectively) were consistent regarding the differences found for the following indicators: eye exam in the previous year and having all requested tests. In the present study, these absolute and relative measures showed that they could be classified as complementary for the indicators mentioned [33]. Generally, measures of absolute inequality are more easily interpreted, as they show, for example, how much coverage of conducting an examination for DM control should increase to achieve equality, making this measure especially useful for health managers to assist in decision making [14].

In Brazil, from 2011 to 2018, primary care qualification policies expanded access to services, improved infrastructure, and ensured improvements in health teamwork processes [11, 34]. However, as this study has shown, and has also been reported by other authors, there is still a need to expand the qualification of clinical practice and access to specific exams [10, 34, 35]. It should be noted that the discontinuity of programs to promote qualification of primary care in Brazil, the reduction of public resources for health and the impact of the COVID-19 pandemic on the health system and the economy, may have worsened the indicators of care for patients with DM, reduced the quality and comprehensiveness of the care offered in the coming years, increasing even further inequality in care among the population.

## Conclusion

We found that the probability of receiving quality care, based on the evaluated indicators, was higher among more educated individuals. In order to reduce social inequalities, it is important that, health services, especially PHC, are organized and the work processes are geared to health needs of the population to which they are targeted at. As a result, health services will be able to progress in the care provided to the poorest, promoting greater equity in health.

## Acknowledgments

Our thanks to the people who were interviewed and contributed to this assessment and to the Brazilian Ministry of Health for making the data openly available.

## Author Contributions

**Conceptualization:** Rosália Garcia Neves, Mirelle de Oliveira Saes, Suele Manjourany Silva Duro, Thaynã Ramos Flores, Elaine Tomasi.

**Formal analysis:** Rosália Garcia Neves, Mirelle de Oliveira Saes, Suele Manjourany Silva Duro, Thaynã Ramos Flores, Elaine Tomasi.

**Writing – review & editing:** Rosália Garcia Neves, Mirelle de Oliveira Saes, Suele Manjourany Silva Duro, Thaynã Ramos Flores, Elaine Tomasi.

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
