## [Decision Letter · Decision Letter 0]

13 Jul 2021

PONE-D-21-11633

Inequalities in care for the people with diabetes in Brazil: a nationwide study, 2019

PLOS ONE

Dear Dr. Neves,

Thank you for submitting your manuscript to PLOS ONE. After careful consideration, we feel that it has merit but does not fully meet PLOS ONE’s publication criteria as it currently stands. Therefore, we invite you to submit a revised version of the manuscript that addresses the points raised during the review process.

As the Academic Editor, I have sent your manuscript for revision to several persons. Currently one expert in the field has already provided the peer-review with very useful comments, suggesting Major Revision. Other experts invited so far have declined the invitation to review the manuscript (it is rather widespread situation, related to different factors).

I would like to save your time, and provide the possibility to perform the revision of the manuscript based on the feedback from one expert in the field, and make it stronger. Once you submit the revised manuscript, I will send it to the expert who provided the feedback, and also involve another reviewer. If you have any objections for this approach, please let me know. If you agree, please proceed this way and submit the revised manuscript along with the point-by-point responses to the reviewer. 

We look forward to receiving your revised manuscript.

Kind regards,

Boris Bikbov

Academic Editor

PLOS ONE

Journal Requirements:

2. Please include your tables as part of your main manuscript and remove the individual files. Please note that supplementary tables (should remain/ be uploaded) as separate "supporting information" files

Additional Editor Comments (if provided):

Reviewers' comments:

Reviewer's Responses to Questions

**Comments to the Author**

1. Is the manuscript technically sound, and do the data support the conclusions?

Reviewer #1: Yes

2. Has the statistical analysis been performed appropriately and rigorously? 

Reviewer #1: Yes

3. Have the authors made all data underlying the findings in their manuscript fully available?

Reviewer #1: Yes

4. Is the manuscript presented in an intelligible fashion and written in standard English?

Reviewer #1: No

5. Review Comments to the Author

Reviewer #1: This is an interesting paper on how social inequalities are translated into health outcomes. My main concerns are: 1. English language must be revised. 2. Methodology needs to be improved with a better explanation on the estimation of both indices: the slope index of inequality (SII) and the concentration index (CIX) and why they are appropriate for this specific study. 3. Discussion needs to discuss Brazil results in the context of regional inequalities in Latin America. 5. The quality and format of figures needs to be improved.

6. PLOS authors have the option to publish the peer review history of their article (what does this mean?). If published, this will include your full peer review and any attached files.

Reviewer #1: **Yes: **Carolina Santamaría-Ulloa

---

## [Author Response · Author response to Decision Letter 0]

31 Aug 2021

August 20th, 2021 

To the Journal Plos One

Subject: answer to editors and reviewers

Dear PLOS ONE Editor 

We thank you for your careful review of the manuscript entitled “Inequalities in care for the people with diabetes in Brazil: a nationwide study, 2019”. We would like to point out that all requests have been met. Below are the details of each one and their respective answer. 

1) English language must be revised. 

Answer: the English language has been revised throughout the manuscript.

2) Methodology needs to be improved with a better explanation on the estimation of both indices: the slope index of inequality (SII) and the concentration index (CIX) and why they are appropriate for this specific study. 

Answer: in the penultimate paragraph of the methodology, the requests, the references that use the indexes in the data analysis and the methodological article that explains in detail how to calculate each one of them have been included. 

3) Discussion needs to discuss Brazil results in the context of regional inequalities in Latin America. 

Answer: after searching the literature, few studies carried out in Latin America dealing with a similar theme were found, however some comparisons have been made with the references we did find. They have been included in paragraphs one and seven of the discussion.

4) The quality and format of figures needs to be improved.

Answer: Figure 2 has been saved in WMF (created in STATA 15.1) format as requested by the journal and Figure 1 in JPG (created in excel).

---

## [Decision Letter · Decision Letter 1]

9 Dec 2021

PONE-D-21-11633R1Inequalities in care for the people with diabetes in Brazil: a nationwide study, 2019PLOS ONE

Dear Dr. Neves,

Thank you for submitting your manuscript to PLOS ONE. After careful consideration, we feel that it has merit but does not fully meet PLOS ONE’s publication criteria as it currently stands. Therefore, we invite you to submit a revised version of the manuscript that addresses the points raised during the review process.

First of all, congratulations to the analysis you have performed. It is especially important considering the hard economic conditions and deterioration of the social support in face of crisis. Please pay attention to the excellent points raised by the reviewers. In addition to this, please consider the following:

Major issues:

1. Please describe in brief the health care system in Brazil, whether persons with diabetes have to pay for medical visits, examination and treatment, or these expenses are covered by the state? Who pay for glucose strips? Whether persons with higher education have health insurance from employer more frequent that those with only primary education? What is the proportion of out-of-pocket payments? How these differences could explain your findings? What is the role of community health centres, if any?

2. You have mentioned that participants have been asked about administration of urine analysis, and indeed kidney disease is one of the major diabetes complications. You have provided (figure 2) very interesting data indicating that the differences in performing urine analysis were not so prominent between education groups compared to the differences in feet or eye examination, or HbA1c evaluation. How it is possible to explain?

3. At line 222 please indicate how to interpret the "general PHC scores".

4. It would be an advantage if you briefly describe how the people reported diabetes but not referred to a physician in a previous 12 months (and excluded from the analysis) differ from those reported diabetes and visited a physician.

5. Please indicate what are the differences between High and higher education - that are among the categories you used in the analysis.

6. The Table 2 will be much more easy to percept if you highlight in bold values with 95%CI higher than 1. Moreover, please use more clear column header and instead of putting "*PR: Prevalence Ratio" in the footnote provide the explanatory column header.

7. Please correct the tile for the Table 3 to make it more readable and avoid repetition of "concentration and". Please explain or comment in the "Discussion" why Slope index of inequality could be rather similar and in the same time concentration index could be rather different, for example considering the "Urine analysis", "Cholesterol or triglycerides" and "Feet examined in the preceding year".

8. Please indicate whether the National Health Survey data are accessible and open.

Minor:

1. Please revise the phrase "We performed all analyses using STATA® 15.0 statistical package, considering the sample design."

2. In the phrase "Advice on practicing physical activity and not drinking too much..." please specify whether drinking alcohol is considered or whatever.

3. Please revise the phrases:

- "The greater the specificity of care, the greater the difference between schooling quintiles, with emphasis on the most educated."

- "The use of schooling in the present study can strengthen our findings..."

4. Please check consistency of English use in the table ("anos ou mais")

5. Please use "95% CI" not "CI 95%"

First of all, congratulations to the analysis you have performed. It is especially important considering the hard economic conditions and deterioration of the social support in face of crisis. Please pay attention to the excellent points raised by the reviewers. In addition to this, please consider the following:

Major issues:

1. Please describe in brief the health care system in Brazil, whether persons with diabetes have to pay for medical visits, examination and treatment, or these expenses are covered by the state? Who pay for glucose strips? Whether persons with higher education have health insurance from employer more frequent that those with only primary education? What is the proportion of out-of-pocket payments? How these differences could explain your findings? What is the role of community health centres, if any?

2. You have mentioned that participants have been asked about administration of urine analysis, and indeed kidney disease is one of the major diabetes complications. You have provided (figure 2) very interesting data indicating that the differences in performing urine analysis were not so prominent between education groups compared to the differences in feet or eye examination, or HbA1c evaluation. How it is possible to explain?

3. At line 222 please indicate how to interpret the "general PHC scores".

4. It would be an advantage if you briefly describe how the people reported diabetes but not referred to a physician in a previous 12 months (and excluded from the analysis) differ from those reported diabetes and visited a physician.

5. Please indicate what are the differences between High and higher education - that are among the categories you used in the analysis.

6. The Table 2 will be much more easy to percept if you highlight in bold values with 95%CI higher than 1. Moreover, please use more clear column header and instead of putting "*PR: Prevalence Ratio" in the footnote provide the explanatory column header.

7. Please correct the tile for the Table 3 to make it more readable and avoid repetition of "concentration and". Please explain or comment in the "Discussion" why Slope index of inequality could be rather similar and in the same time concentration index could be rather different, for example considering the "Urine analysis", "Cholesterol or triglycerides" and "Feet examined in the preceding year".

8. Please indicate whether the National Health Survey data are accessible and open.

Minor:

1. Please revise the phrase "We performed all analyses using STATA® 15.0 statistical package, considering the sample design."

2. In the phrase "Advice on practicing physical activity and not drinking too much..." please specify whether drinking alcohol is considered or whatever.

3. Please revise the phrases:

- "The greater the specificity of care, the greater the difference between schooling quintiles, with emphasis on the most educated."

- "The use of schooling in the present study can strengthen our findings..."

4. Please check consistency of English use in the table ("anos ou mais").

5. Please use "95% CI" not "CI 95%".

We look forward to receiving your revised manuscript.

Kind regards,

Boris Bikbov

Academic Editor

PLOS ONE

Journal Requirements:

Reviewers' comments:

Reviewer's Responses to Questions

**Comments to the Author**

1. If the authors have adequately addressed your comments raised in a previous round of review and you feel that this manuscript is now acceptable for publication, you may indicate that here to bypass the “Comments to the Author” section, enter your conflict of interest statement in the “Confidential to Editor” section, and submit your "Accept" recommendation.

Reviewer #1: (No Response)

Reviewer #2: (No Response)

2. Is the manuscript technically sound, and do the data support the conclusions?

Reviewer #1: Yes

Reviewer #2: Yes

3. Has the statistical analysis been performed appropriately and rigorously? 

Reviewer #1: Yes

Reviewer #2: Yes

4. Have the authors made all data underlying the findings in their manuscript fully available?

Reviewer #1: Yes

Reviewer #2: Yes

5. Is the manuscript presented in an intelligible fashion and written in standard English?

Reviewer #1: No

Reviewer #2: Yes

6. Review Comments to the Author

Reviewer #1: This paper is not written in standard English. Although the authors declare they had it revised, it still needs an English revision.

Reviewer #2: This paper is a revision (after a first review?) describing educational inequalities in diabetes care in Brazil, based on the PNS 20219. Similar descriptions have been made earlier by the same author based on the PNS 2013. The paper shows however no comparisons over time for the inequalities in diabetes care. There are data from all states in in Brazil, states that might differ considerably, both in socioeconomic terms and in possibly also in in terms of policy and diabetes care. Quantitative comparison over time and between states/regions would have been very informative and policy relevant, and would heighten the interest for these data.

The analysis is adjusted for age, region and self- reported skin colour .Figure 1 and table shows essentially the same information as is shown in table 3. Tables illustrating inequalities stratified on age and state/region and comparisons over time would have been more informative.

7. PLOS authors have the option to publish the peer review history of their article (what does this mean?). If published, this will include your full peer review and any attached files.

Reviewer #1: No

Reviewer #2: No

---

## [Author Response · Author response to Decision Letter 1]

28 Feb 2022

Dear reviewers and editor

We greatly appreciate your comments to improve the manuscript. Below is information about your considerations.

- This manuscript has undergone two English revisions. It would be very helpful for us to point out which points need to be reviewed, as we were unable to identify them.

- The article referred to with data from 2013 was only with a sample of elderly people, aged 60 years or older, unlike the one that used a sample of adults over 18 years of age. In addition, this article presents analyzes that were not performed with the 2013 data, such as the analysis adjusted through Poisson regression, in which we can identify the magnitude of the effect measure in the different education categories.

- And regarding the stratification by age, region, state, we chose not to explore it in this article, since, as it is an incipient topic in the literature, we chose to carry out a national analysis, but we intend to explore the data in future studies.

---

## [Decision Letter · Decision Letter 2]

30 Mar 2022

PONE-D-21-11633R2Inequalities in care for the people with diabetes in Brazil: a nationwide study, 2019PLOS ONE

Dear Dr. Neves,

Thank you for submitting your manuscript to PLOS ONE. After careful consideration, we feel that it has merit but does not fully meet PLOS ONE’s publication criteria as it currently stands. Therefore, we invite you to submit a revised version of the manuscript that addresses the points raised during the review process. Please take attention to the very useful comments of the Reviewer bout the need to provide more details about the health care system in Brazil.One of the major focus of the analysis is SII and CIX evaluation, and they are described in Methods and Results. The Discussion the inequalities described mainly without mentioning these indexes, and even lines 278-285 that refer to these indexes discussed more in general than in details. When you describe in the Discussion "Absolute inequality and relative inequality", please add "(expressed by SII and CIX, resp.)". It would be better if you put in the "Discussion" some highlights and explanations, along with the public health recommendations, based on your analysis of these indexes. For example, in the responses to reviewers you explained why these indexes had prominent difference between urianalysis and eye examination, and this could be highlighted also in the "Discussion" in a more explicit way.Please check and correct some phrases that could be improved for the better interpretation. For example, the phrase "All requested tests were found in less than half of the sample." could be more clear if transformed to something like "Less than a half of studied population had performed all recommended tests during the 12 month period." Please note that such improvement could be made for several phrases in different parts of the text.

 Please submit your revised manuscript by May 14 2022 11:59PM. If you will need more time than this to complete your revisions, please reply to this message or contact the journal office at plosone@plos.org. Please include the following items when submitting your revised manuscript:A rebuttal letter that responds to each point raised by the academic editor and reviewer(s). You should upload this letter as a separate file labeled 'Response to Reviewers'.A marked-up copy of your manuscript that highlights changes made to the original version. You should upload this as a separate file labeled 'Revised Manuscript with Track Changes'.An unmarked version of your revised paper without tracked changes. You should upload this as a separate file labeled 'Manuscript'.If applicable, we recommend that you deposit your laboratory protocols in protocols.io to enhance the reproducibility of your results. Protocols.io assigns your protocol its own identifier (DOI) so that it can be cited independently in the future. For instructions see: https://journals.plos.org/plosone/s/submission-guidelines#loc-laboratory-protocols. Additionally, PLOS ONE offers an option for publishing peer-reviewed Lab Protocol articles, which describe protocols hosted on protocols.io. Read more information on sharing protocols at https://plos.org/protocols?utm_medium=editorial-email&utm_source=authorletters&utm_campaign=protocols.

We look forward to receiving your revised manuscript.

Kind regards,

Boris Bikbov, MD, PhD

Academic Editor

PLOS ONE

Journal Requirements:

Reviewers' comments:

Reviewer's Responses to Questions

**Comments to the Author**

1. If the authors have adequately addressed your comments raised in a previous round of review and you feel that this manuscript is now acceptable for publication, you may indicate that here to bypass the “Comments to the Author” section, enter your conflict of interest statement in the “Confidential to Editor” section, and submit your "Accept" recommendation.

Reviewer #1: All comments have been addressed

Reviewer #2: All comments have been addressed

2. Is the manuscript technically sound, and do the data support the conclusions?

Reviewer #1: (No Response)

Reviewer #2: Yes

3. Has the statistical analysis been performed appropriately and rigorously? 

Reviewer #1: (No Response)

Reviewer #2: Yes

4. Have the authors made all data underlying the findings in their manuscript fully available?

Reviewer #1: (No Response)

Reviewer #2: Yes

5. Is the manuscript presented in an intelligible fashion and written in standard English?

Reviewer #1: (No Response)

Reviewer #2: Yes

6. Review Comments to the Author

Reviewer #1: The authors have adequately addressed my comments. I recommend to accept this manuscript for publication.

Reviewer #2: It is well written paper with 2019 data describing inequalities in DM care. I have only two comments:

From the description of the Brazilian context on page 3 people could get the impression that SUS is the only health care system. According to my knowledge more than 25% of the population has an alternative health insurance paid privately or by the employer, that give them access to the huge private sector within the Brazilian health sector. I assume that access to private sector is much higher among well educated, and therefore explains a large part of the inequalities in care. This context needs to be described and discussed in the paper. It would have been interesting to look at the SII for different federal units, since that would tell us whether different state policies make a difference.

On page 11 line 271-275 it is indicated that education is used as proxy for economic level. Why? Education is often used as a measure of socioeconomic position in society. I agree that income or wealth could be very interesting measures but they might not be accessible to the authors. Those lines can be omitted.

7. PLOS authors have the option to publish the peer review history of their article (what does this mean?). If published, this will include your full peer review and any attached files.

Reviewer #1: **Yes: **Carolina Santamaria-Ulloa

Reviewer #2: **Yes: **Finn Diderichsen

---

## [Author Response · Author response to Decision Letter 2]

13 May 2022

May 02th, 2022 

To the Journal Plos One

Subject: answer to editors and reviewers

Dear PLOS ONE Editor 

We thank you for your careful review of the manuscript entitled “Inequalities in care for the people with diabetes in Brazil: a nationwide study, 2019”. We would like to point out that all requests have been met. Below are the details of each one and their respective answer. 

• Please take attention to the very useful comments of the Reviewer bout the need to provide more details about the health care system in Brazil.

Answer: done.

• One of the major focus of the analysis is SII and CIX evaluation, and they are described in Methods and Results. The Discussion the inequalities described mainly without mentioning these indexes, and even lines 278-285 that refer to these indexes discussed more in general than in details. When you describe in the Discussion "Absolute inequality and relative inequality", please add "(expressed by SII and CIX, resp.)". It would be better if you put in the "Discussion" some highlights and explanations, along with the public health recommendations, based on your analysis of these indexes. For example, in the responses to reviewers you explained why these indexes had prominent difference between urianalysis and eye examination, and this could be highlighted also in the "Discussion" in a more explicit way.

Answer: thanks for the comment. The greatest inequalities according to the indexes were found in the examinations of feet and eyes. In the discussion, the possibility of the lack of materials necessary for this evaluation in primary health care units was commented, leading to a low prevalence, especially in people with lower socioeconomic status, who are the ones who most use these services. Paragraph 262 to 267.

• Please check and correct some phrases that could be improved for the better interpretation. For example, the phrase "All requested tests were found in less than half of the sample." could be more clear if transformed to something like "Less than a half of studied population had performed all recommended tests during the 12 month period." Please note that such improvement could be made for several phrases in different parts of the text.

Answer: done.

• From the description of the Brazilian context on page 3 people could get the impression that SUS is the only health care system. According to my knowledge more than 25% of the population has an alternative health insurance paid privately or by the employer, that give them access to the huge private sector within the Brazilian health sector. I assume that access to private sector is much higher among well educated, and therefore explains a large part of the inequalities in care. This context needs to be described and discussed in the paper.

Answer: thanks for the suggestion. We write about the topic on lines 77 to 80.

• It would have been interesting to look at the SII for different federal units, since that would tell us whether different state policies make a difference.

Answer: thanks for the comment, it's an excellent suggestion. We intend to work on future studies that investigate inequalities by federation units. For the present study we preferred to use a nationally representative sample to investigate inequalities at the individual level using a socioeconomic variable.

• On page 11 line 271-275 it is indicated that education is used as proxy for economic level. Why? Education is often used as a measure of socioeconomic position in society. I agree that income or wealth could be very interesting measures but they might not be accessible to the authors. Those lines can be omitted.

Answer: thanks for the suggestion. We agree with you and changed to socioeconomic level

---

## [Decision Letter · Decision Letter 3]

3 Jun 2022

Inequalities in care for the people with diabetes in Brazil: a nationwide study, 2019

PONE-D-21-11633R3

Dear Dr. Neves,

We’re pleased to inform you that your manuscript has been judged scientifically suitable for publication and will be formally accepted for publication once it meets all outstanding technical requirements.

Kind regards,

Boris Bikbov, MD, PhD

Academic Editor

PLOS ONE

Additional Editor Comments (optional):

Reviewers' comments:

Reviewer's Responses to Questions

**Comments to the Author**

1. If the authors have adequately addressed your comments raised in a previous round of review and you feel that this manuscript is now acceptable for publication, you may indicate that here to bypass the “Comments to the Author” section, enter your conflict of interest statement in the “Confidential to Editor” section, and submit your "Accept" recommendation.

Reviewer #2: All comments have been addressed

2. Is the manuscript technically sound, and do the data support the conclusions?

Reviewer #2: Yes

3. Has the statistical analysis been performed appropriately and rigorously? 

Reviewer #2: Yes

4. Have the authors made all data underlying the findings in their manuscript fully available?

Reviewer #2: Yes

5. Is the manuscript presented in an intelligible fashion and written in standard English?

Reviewer #2: Yes

6. Review Comments to the Author

Reviewer #2: The authors have shortly adressed my comments.

7. PLOS authors have the option to publish the peer review history of their article (what does this mean?). If published, this will include your full peer review and any attached files.

Reviewer #2: No

---

## [Editor Report · Acceptance letter]

21 Jun 2022

PONE-D-21-11633R3 

Inequalities in care for the people with diabetes in Brazil: a nationwide study, 2019 

Dear Dr. Neves:

I'm pleased to inform you that your manuscript has been deemed suitable for publication in PLOS ONE. Congratulations! Your manuscript is now with our production department. 

Kind regards, 

on behalf of

Dr. Boris Bikbov 

Academic Editor

PLOS ONE